

# Evaluation of wind farm parameterizations in the WRF model under different atmospheric stability conditions with high-resolution wake simulations

Oscar García-Santiago[1], Andrea N. Hahmann[1], Jake Badger[1], and Alfredo Peña[1]

[1]Department of Wind and Energy Systems, Technical University of Denmark, Roskilde, Denmark

**Correspondence:** Oscar García-Santiago (osmasa@dtu.dk)

**Abstract.**

Wind farm parameterizations (WFPs) are used in mesoscale models for predicting wind farm power production and its impact on wind resources while considering the variability of the regional wind climate. However, the performance of WFPs is influenced by various factors including atmospheric stability. In this study, we compared two widely used WFPs in the Weather
Research and Forecasting (WRF) model to large-eddy simulations (LES) of turbine wakes performed with the same model. The Fitch scheme and the Explicit Wake Parameterization were evaluated for their ability to represent wind speed and turbulent kinetic energy (TKE) in a two-turbine wind farm layout under neutral, unstable, and stable atmospheric stability conditions. To ensure a fair comparison, the inflow conditions were kept as close as possible between the LES and mesoscale simulations for each type of stability condition, and the LES results were spatially aggregated to align with the mesoscale grid spacing. Our
findings indicate that the performance of WFPs varies depending on the specific variable (wind speed or TKE) and the area of interest downwind of the turbine when compared to the LES reference. The WFPs can accurately depict the vertical profiles of the wind speed deficit for either the grid cell containing the wind turbines or the grid cells in the far wake, but not both simultaneously. The WFPs with an explicit source of TKE overestimate TKE values at the first grid cell containing the wind turbine; however, for downwind grid cells, agreement improves. On the other hand, WFPs without a TKE source underestimate
TKE in all downwind grid cells. These agreement patterns between the WFPs and the LES reference are consistent under the three atmospheric stability conditions. However, the WFPs resemble less the wind speed and TKE from the LES reference under stable conditions than under neutral or unstable conditions.

## 1 Introduction

The growing need for clean and sustainable energy sources has led to a demand for large-scale and potentially concentrated
deployment of wind farms (Ministry of Climate and Energy, 2022, 2023). Mesoscale models represent the atmospheric state and its dynamics and can now include the effect of wind farms and wind farm clusters on the atmospheric flow. These models are being used in the wind farm planning phase, as they consider the potential impacts of a proposed farm or a cluster of wind farms on the surrounding wind energy resources, together with the regional climate of the site (Lundquist et al., 2019; Akhtar





et al., 2021; Pryor et al., 2022; Fischereit et al., 2022b). Given the economical, political and environmental significance of these
applications, it is essential that these models accurately represent turbine-atmosphere interactions.

The Weather Research and Forecasting (WRF) model (Skamarock et al., 2021) is a mesoscale model that has become an essential tool for wind resource assessments (Olsen et al., 2017), as reflected in its use for the creation of modern wind atlases (Hahmann et al., 2020; Bodini et al., 2020; Davis et al., 2023). The WRF model can incorporate the effect of wind farms through a wind farm parameterization (WFP). Several WFPs have been implemented, with the Fitch scheme (Fitch et al., 2012), hereafter referred to as Fitch, and the explicit wake parameterization (EWP; Volker et al., 2015) being the two most commonly used (Fischereit et al., 2022a). Both WFPs have been extensively discussed in the literature, and research findings are usually framed in comparisons of wind speed-based wake extension, turbulent kinetic energy (TKE), and power production (Pryor et al., 2020; Shepherd et al., 2020; Larsén and Fischereit, 2021; Ali et al., 2023). Across these studies, a common finding is that the Fitch scheme typically simulates deeper wakes and higher TKE than the EWP scheme, the latter often underestimating TKE compared to limited measurements (Larsén and Fischereit, 2021).

The differences in outcomes from these WFPs rely on their methods to parameterise the impact of wind turbines on the flow. The two schemes differ in how they vertically distribute the wind speed deficit and whether they include an explicit source of TKE. Each WFP relies on a single controlling parameter that impacts one of the behaviours of the WFP outlined above. In the Fitch scheme, the correction factor $c_f$, initially implemented as a bug fix by Archer et al. (2020), modulates the TKE production from the turbines. The EWP scheme, without an explicit TKE source, defines an initial length scale ($\sigma_0$) that partially controls how the velocity deficit is vertically distributed (Volker et al., 2015). Understanding the WFPs, their controlling parameters, and their interactions with the modelled atmospheric conditions is crucial for their effective use and future improvements.

The importance of validating the WFPs under different atmospheric conditions cannot be overstated, as the state of the atmosphere impacts the wind farm's performance. Atmospheric stability is known to influence both wake behaviour and turbine-induced turbulence (Porté-Agel et al., 2020). However, validating these two aspects for WFPs poses considerable challenges under realistic conditions (Fischereit et al., 2022a). A primary issue is the scarcity of extensive, accessible, and long-term datasets for comparison. This often leads to WFPs that are validated within brief atmospheric episodes representing a limited range of conditions. Furthermore, validating the WFPs against measurements greatly relies on the mesoscale model's ability to accurately represent the background conditions (Lee and Lundquist, 2017; Fischereit et al., 2022a). Any inaccuracies in this representation can significantly impact the performance of the WFPs, introducing an additional layer of complexity to the validation process.

In response to these validation challenges, some studies evaluate the WFPs results against high-fidelity wake simulations, e.g., from large-eddy simulations (LES; Eriksson et al., 2015; Vanderwende et al., 2016; Archer et al., 2020; Peña et al., 2022). LES, using actuator discs (ADs) or actuator lines methods, can provide a high-resolution and detailed representation of wind turbine wakes with a high computational cost. In particular, the WRF model can also operate in LES mode with the option of using ADs (Mirocha et al., 2014; Kale et al., 2022) and has been used to study non-neutral boundary layers (Kosović and Curry, 2000; Mirocha et al., 2018; Simon et al., 2019; Peña et al., 2021; Wu et al., 2023). These LES methods, including WRF-LES



with ADs, can improve the accuracy of WFPs by providing a detailed reference for their refinement (Abkar and Porté-Agel, 2015; Pan and Archer, 2018; Archer et al., 2020).

The Fitch and EWP schemes and their recommended parameter values have been calibrated under neutral atmospheric conditions using LES representations (Archer et al., 2020) or averaged stability-filtered measurements (Volker et al., 2015). However, the accuracy of these WFPs under non-neutral atmospheric stability conditions has not been clearly determined and requires substantial exploration. Therefore, our study's primary objective is to deepen our understanding of WFPs in mesoscale models by evaluating them against LES using ADs under different atmospheric stability conditions. Specifically,

our study seeks to answer two key research questions: (1) How comparable are mesoscale inflow conditions to those simulated by LES under different atmospheric stabilities before adding wind turbines? and (2) How accurately can the EWP and Fitch WFPs schemes represent the impact of wind turbines on the flow across different atmospheric stability conditions compared to an analogous LES-AD-based simulation?

To address these questions, our study conducts controlled simulations that characterise the structure of neutral, unstable,

and stable atmospheric boundary layers (ABLs). We focus on the wake effects under these atmospheric conditions while maintaining, as close as possible, the inflow conditions in the LES and mesoscale simulations. More importantly, the LES and mesoscale runs are performed with the same version of the WRF model, thus ensuring consistency in both physics and numerical approximations between the two modelling approaches.

This study is structured as follows. Section 2 presents the configurations of the WRF model needed for the LES and

mesoscale frameworks to represent the three atmospheric stability conditions. This section also details the WFPs, the AD model, and the methods used for comparison. Section 3 evaluates the similarities between the LES and mesoscale inflow conditions. Section 4 details our wind turbine simulation results. Finally, Sections 5 and 6 discuss our results and summarise the key conclusions of the study.

## 2   Methods

We perform a series of WRF model simulations at different time and spatial scales to verify the behaviour of the WFPs (Fitch and EWP) under different atmospheric stability conditions and WFP-specific parameter values. These simulations are divided into coarse resolution runs using PBL schemes (i.e., mesoscale, from now on referred to as PBL simulations), and high-resolution runs (i.e., LES) as reference. We first simulate the atmospheric flow for three distinct atmospheric stability regimes with LES without turbines. Then, we try to match the LES inflow conditions as closely as possible in the PBL simulations.

Afterwards, we apply the WFPs, with different WFP-specific parameter values, in the PBL simulations and the AD in the LES.

The two modelling approaches use the same version of the WRF model (version 4.2.2), which includes an implementation of an AD model (Mirocha et al., 2014) and two WFPs schemes: Fitch (Fitch et al., 2012) and EWP (Volker et al., 2015). We investigate two simple turbine layouts: one with an isolated turbine and the second with two flow-aligned turbines with a separation of seven rotor diameters ($D$). The two layouts use the DTU 10-MW reference turbine (Bak et al., 2013) with a hub

height of 119 m and $D = 178.3$ m.





In the following subsections, we present the configurations for the WRF model simulations, the details used for developing the different atmospheric boundary layers in the LES and PBL simulations, the parameter values used in the WFPs, and finally, the skill metric used in the assessments.

## 2.1 The WRF model setup

We run the WRF model in idealised mode (no real forcing but prescribed initial values of vertical profiles of potential temperature and horizontal velocity components) with minimal physical parameterizations to isolate the development of the atmospheric boundary layer within the LES and PBL frameworks. The two frameworks use two computational domains: an outer and an inner domain, the latter at the centre of the former. The boundary conditions for these domains are periodic for the outer and nested for the inner. Both modelling approaches share the same vertical distribution of 120 model levels, but differ horizontally in both the number of grid points and grid spacing. The grid aspect ratio is 1:3 in both frameworks. We use the same surface-layer scheme in which Monin–Obukhov similarity theory is applied. Additional configuration settings and parameters common to both modelling approaches are detailed in Table 1.

**Table 1.** Common configuration settings for the LES and PBL frameworks in the WRF model.

| Common setup | Setting |
| --- | --- |
| Vertical discretization | 120 vertical levels with model top at 2 km |
|  | First lowest 40 model levels with a spacing of 5 m |
| Damping layer | Rayleigh damping for the upper 400-m layer |
| Horizontal grid | Two domains with a grid aspect ratio of 1:3 |
| Nesting | One-way nesting |
| Lateral boundary conditions | Periodic for the outer and nested for the inner domains |
| Surface layer scheme | Revised Monin-Obukhov scheme |
|  | (`sf_sfclay_physics = 1`; Jiménez et al., 2012) |
| Surface roughness length | $2 \times 10^{-4}$ m (offshore conditions) |
| Coriolis parameter | $1.2 \times 10^{-4}$ s$^{-1}$ |

We develop neutral, unstable, and stable boundary layers in both modelling approaches (LES and PBL) following the procedures of Mirocha et al. (2018) and Peña et al. (2021) for the LES and Rybchuk et al. (2022) for the PBL frameworks. The method consists of adjusting the initial vertical structure of the ABL and applying constant surface heat fluxes (or heating rates) through the model's surface-layer scheme. The model is then run with boundary conditions until the desired wind profile and thermodynamic structure is achieved. All simulations are initiated with a vertically uniform geostrophic wind. The initial vertical profile of potential temperature is constant (290 K) from the surface up to the inversion, whose location and strength vary for each type of ABL and framework. The value of surface heat flux (or heating rate) depends on the strength of the desired stability condition and the framework.





## 2.2 LES specifics

The LES framework employs a horizontal domain configuration detailed in Fig. 1 and Table 2. We use the subgrid-scale model by Deardorff (1980), integrating a prognostic equation of the subgrid TKE. The choice of inversion height, inversion strength, and surface fluxes is based on the studies from Mirocha et al. (2018), Peña et al. (2021), and Kale et al. (2022), but for offshore
115   conditions. Table 3 summarises these values used in the LESs for the three atmospheric conditions. The table also shows the adjusted initial geostrophic wind components to obtain a wind speed of $10\ \mathrm{m\ s^{-1}}$ and a wind direction of $270°$ at hub height.

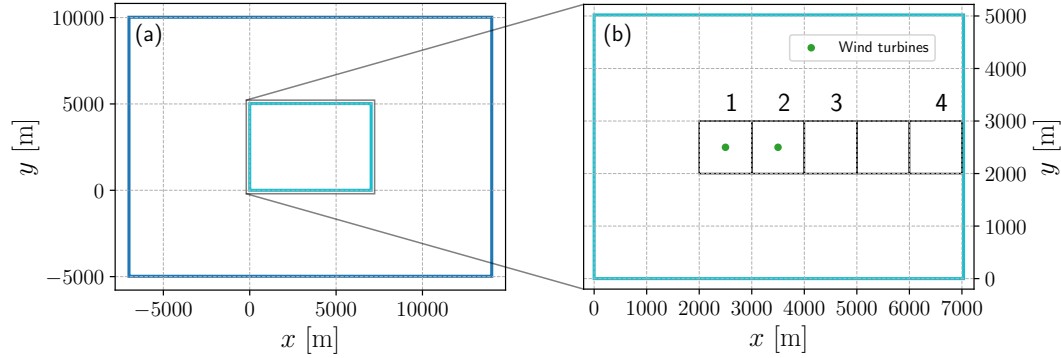

**Figure 1.** (a) Outer and (b) inner domain configuration in the LES framework. The black numerated squares in the inner domain indicate the areas where the LES outputs are spatially averaged. The domain boundaries are shown in blue, and the green dots indicate the position of the turbines when activated.

**Table 2.** Specific WRF model configuration for the LES and PBL frameworks.

| Framework | Domain | Grid points $(x \times y)$ | Horizontal grid spacing [m] | Time step [s] | Output frequency [s] | Turbulence scheme |
|---|---|---|---|---|---|---|
| LES | Inner | $703 \times 502$ | 10 | 0.2 | 10 | Subgrid-scale model of |
|  | Outer | $700 \times 500$ | 30 | 0.6 | - | Deardorff (1980) |
| PBL | Inner | $163 \times 43$ | 1000 | 10 | 600 | MYNN PBL scheme, |
|  | Outer | $160 \times 40$ | 3000 | 30 | - | (Nakanishi and Niino, 2009) |

We develop turbulence and the ABL in the outer domain with periodic lateral boundary conditions and initial random perturbations in the potential temperature field, which vary within a $\pm 0.5$ K (Mirocha et al., 2014). Considering the inherent unsteadiness of LES, the mean flow is considered "steady" when the change in the standard deviation of hub-height wind speed and direction, and the friction velocity values are less than $2\%$ at the last hour of simulation. Note that the boundary layer might





be continuously growing. Nearly steady conditions and boundary layer height values typical of the offshore atmosphere are reached in the outer domain of the LES framework after 10 h, 8 h, and 16 h for neutral, unstable, and stable conditions, respectively (see Table 3).

**Table 3.** Overview of the initial profile values, surface heat fluxes or heating rates, and total simulation time used to obtain the vertical structure of the three ABLs using LES and PBL frameworks in the WRF model. All parameters are chosen to obtain a wind speed of 10 m s$^{-1}$ and a wind direction of 270° at hub height (119 m) for all the cases.

| Framework | Atmospheric stability | Geostrophic wind $G_x$, $G_y$ [m s$^{-1}$] | $\overline{w'\theta'}$ [K m s$^{-1}$] | Heating rate [K h$^{-1}$] | Inversion height [m] | Inversion strength [K km$^{-1}$] | Spin-up time [h] | Total run time [h] |
|---|---|---|---|---|---|---|---|---|
| LES | Neutral | 10.2002, −2.0381 | – | – | 700 | 10 | 10 | 12 |
| | Unstable | 9.9106, −1.6544 | 0.02 | – | 700 | 4 | 8 | 10 |
| | Stable | 9.6709, −2.0735 | – | −0.06 | 230 | 10 | 16 | 18 |
| PBL | Neutral | 10.7128, −1.4229 | – | – | 700 | 10 | 10 | 16 |
| | Unstable | 10.3529, −1.1004 | – | 0.15 | 1100 | 4 | 8 | 14 |
| | Stable | 9.7089, −1.6185 | – | −0.10 | 230 | 10 | 10 | 16 |

After the near-steady stated is reached in the outer domain, the inner domain is introduced and run for two hours while keeping the periodic boundary conditions in the outer domain. We use an AD to simulate the turbine's impact on the flow in the innermost LES domain. The AD integrates turbine-induced (1D momentum theory) and local aerodynamic forces (blade element theory) through the blade element momentum theory (Mikkelsen, 2004; Mirocha et al., 2014). Our AD implementation uses the tip correction factors from Shen et al. (2005), which Peña et al. (2022) showed provided good agreement with the normal and tangential forces along the blade from Bak et al. (2013). Also Peña et al. (2022) showed good agreement with the integrated power and thrust curves compared to those of Bak et al. (2013). The AD's yaw controller is also deactivated to compare the results more fairly with those of the WFPs.

### 2.3 Configuration of the PBL simulations and wind farm parameterizations

We use the Mellor-Yamada Nakanishi-Niino (MYNN; Nakanishi and Niino, 2009) 1.5-order scheme to model vertical turbulence exchange processes in the ABL. The MYNN scheme prognoses the TKE, and the horizontal advection of TKE is activated. The specific domain configurations for the PBL framework are summarised in Table 2. We adjust the initial vertical structure of the ABL (i.e., geostrophic wind and inversion height, and strength values), the surface fluxes (or heating rate), and the total simulation time in the PBL framework to match the hub-height wind speed and direction, and the atmospheric stability condition to those obtained with the LES. Table 3 shows the adjusted values used to create the three ABLs in the PBL framework. In addition, for neutral and unstable ABLs, we set a capping inversion of 0.02 K m$^{-1}$ in the 100 m layer above the



location of the inversion height to inhibit boundary growth. Then, the inversion strength above the capping inversion is kept as listed in Table 3.

For each ABL type, and similar to the LES cases, the model is run until friction velocity and the hub-height wind components are steady. After that, we introduce and run the inner domain for six hours while preserving the periodic conditions in the outer domain. The total simulation and the spin-up time needed for each ABL in the PBL framework are summarised in Table 3.

We use the Fitch and EWP schemes to model turbine flow impacts in the innermost PBL domain, with turbine locations centred at each 1 km grid cell. We examine how the parameters of these schemes, the correction factor ($c_f$; Archer et al., 2020) for Fitch and the initial length scale ($\sigma_0$; Volker et al., 2015) for EWP, influence the wind and TKE fields under the three ABL types. For Fitch, we use $c_f = 0.0, 0.5, \text{and} 0.75$ and the recommended value found for a neutral ABL (0.25; Archer et al., 2020). For EWP, we use $\sigma_0 = 0.6r_0, 1.0r_0, \text{and} 1.7r_0$ as suggested by Volker et al. (2015), where $r_0$ denotes the radius of the turbine

rotor. Appendix A presents a detailed explanation of these scheme-specific parameters and their relation to the velocity deficit and TKE. We further summarise the setup of the PBL simulations in Table 4.

**Table 4.** Parameters of the wind turbine modelling schemes. See text and Appendix A for details.

| Framework | Model | Parameter value | Label |
|---|---|---|---|
| PBL | EWP | $\sigma_0 = 0.6r_0$ | EWP-0.6 |
| | | $\sigma_0 = 1.0r_0$ | EWP-1.0 |
| | | $\sigma_0 = 1.7r_0$ (default) | EWP-1.7 |
| | Fitch | $c_f = 0.0$ | Fitch-0.0 |
| | | $c_f = 0.25$ (default) | Fitch-0.25 |
| | | $c_f = 0.5$ | Fitch-0.5 |
| | | $c_f = 0.75$ | Fitch-0.75 |
| LES | AD | – | LES-AD |

## 2.4 Evaluation methods

After running the WRF model and achieving near-steady conditions (see spin-up time from Table 3), we perform two restart simulations within the innermost LES and PBL domains: with and without the influence of turbines. Turbine simulations are

conducted for two turbine layouts introduced earlier in the section: an isolated turbine and two turbines aligned with the main flow. After performing all the simulations, all model results from the innermost domains are time-averaged within the last hour of the simulation (see total run times in Table 3). Following the approaches in Archer et al. (2020) and Peña et al. (2022), we horizontal-spatially average all 10-m horizontal resolution LES results within 1 km $\times$1 km areas to compare with the coarser 1-km horizontal resolution PBL results.





After performing one-hour time averages for the PBL simulations and spatially averaging the LES results, we compare the WFPs' results against the LES-AD results for each of the three ABL types. This involves a comparison of the vertical profiles of velocity and TKE from four averaged areas: Areas 1, 2 that contain wind turbines; Areas 3 and 4 downstream) as illustrated in Fig. 1b. The vertical profiles are the differences ($\Delta$) between simulations with and without turbines. We compare the total TKE in the LES, which adds the subgrid to the resolved contributions, to the TKE values from the PBL. The resolved TKE,

defined as $\overline{k}_{res} = \frac{1}{2}(\overline{u'u'} + \overline{v'v'} + \overline{w'w'})$, is calculated from the LES results using the 10-s values from the last hour of the simulations.

    Lastly, we use the root mean square difference (RMSD) as a skill measure to quantify the differences between two vertical profiles. It is defined as

$$\text{RMSD} = \sqrt{\frac{1}{n}\sum_{i=1}^{n}\left(\Delta\overline{\eta}^i_{\text{PBL}} - \Delta\langle\overline{\eta}\rangle^i_{\text{LES}}\right)^2}, \tag{1}$$

where $n = 32$ is the number of vertical levels intersecting the rotor extension, the overbar denotes the one-hour mean, the angle brackets $\langle\rangle$ the one-kilometre average area, and $\eta$ is the analysed variable at each vertical model level from the PBL and LES runs. Equation (1) is used in the comparison of horizontal wind speed ($U = \sqrt{u^2 + v^2}$), and wind direction ($\phi$). It is also used in simulations without turbines; that is, no $\Delta$ operator is needed.

## 3 Inflow profiles

We analyse the differences between the LES and PBL vertical profiles of wind speed and direction, potential temperature, and TKE obtained for the three ABL types before simulating the turbines. We also compute the boundary-layer height by finding the height of the maximum vertical temperature gradient, and the Obukhov length (WRF model output of the surface-layer scheme) is computed from the surface fluxes. Lastly, the degree of similarity between the PBL and LES inflow vertical profiles of wind speed and direction is quantified with the RMSD using Eq. (1) over the 32 vertical levels intersecting the area where

the wind turbine rotor will be placed later.

    Figure 2 compares the inflow vertical profiles from the time-averaged PBL and the temporal and spatial averaged LES over Area 1 in Fig. 1b. In the neutral case, the vertical profiles of potential temperature in the PBL and LES frameworks (Fig. 2c) closely resemble each other, with boundary-layer heights of 747.2 m for LES and 754.2 m for PBL (Table 5). The TKE is similar between the LES and PBL runs within the upper part of the boundary layer. However, the closer to the surface the

larger the differences, with lower TKE in the PBL than in the LESs (Fig. 2d). The neutral ABL wind speed shows contrarily similar values and shear below hub height and tends to show more differences in the upper part of the domain. This behaviour also applies to the wind direction. Although the PBL height is nearly the same for both LES and PBL, the location of the jet's nose is about 100 m apart.

    In the unstable case, vertical profiles generally agree between the PBL and LES runs, and the largest differences appear in

the upper part of the domain for the wind speed and direction (Fig. 2a and b). Within the rotor area, the PBL and LES results show a RMSD of 0.01 m s$^{-1}$ and 0.6° for the wind speed and direction, respectively (Table 5). Note that the values of the

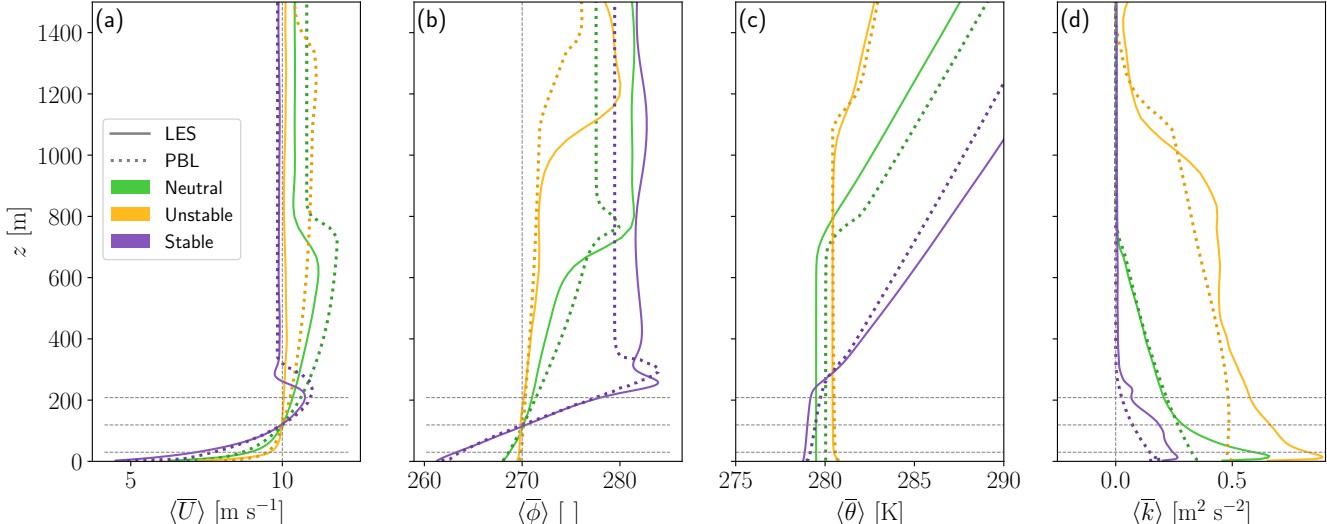

**Figure 2.** Inflow vertical profiles of (a) horizontal wind speed, (b) wind direction, (c) potential temperature, and (d) TKE for the three atmospheric stability cases using the WRF model in PBL (dotted line) and LES (solid lines) frameworks. The top, bottom, and hub height of the wind turbine rotor are depicted with horizontal grey lines for reference.

**Table 5.** Comparison of the ABL statistics between the PBL and LES simulations for the three atmospheric stabilities. Boundary-layer heights and Obukhov lengths along with the RMSD differences of the wind speed and direction within the wind turbine rotor.

| Atmospheric stability | Boundary layer height [m] | | Inverse Obukhov length [m$^{-1}$] | | RMSD wind speed [m s$^{-1}$] | RMSD wind direction [°] |
|---|---|---|---|---|---|---|
| | LES | PBL | LES | PBL | | |
| Neutral | 747.2 | 754.2 | 0.0 | 0.0 | 0.03 | 0.03 |
| Unstable | 1032.0 | 1191.0 | $-5.2 \times 10^{-3}$ | $-6.1 \times 10^{-3}$ | 0.01 | 0.06 |
| Stable | 250.0 | 258.3 | $5.4 \times 10^{-3}$ | $6.8 \times 10^{-3}$ | 0.02 | 0.07 |



inverse Obukhov length (Table 5) are similar between the PBL and LES runs, depicting an unstable atmospheric surface layer. The PBL run shows the largest differences in TKE compared to the LES run in this unstable case.

The stable case, as expected, exhibits a shallower boundary layer than the neutral and unstable cases. The PBL simulation agrees with the LES throughout the boundary layer for wind speed and direction with a RMSD of 0.02 m s$^{-1}$ and 0.07°, respectively. However, the PBL simulation consistently shows less TKE than the LES. Furthermore, the jet's nose in the PBL simulation (at about 250 m) is higher than that in the LES (at about 230 m), which is consistent in the three ABL types.

In summary, the wind speed and direction profiles within the rotor area in the PBL and LES frameworks for the three stability classes match well (Table 5). This demonstrates the comparability between the simulation frameworks in capturing key features of the inflow conditions.

## 4 Simulating wind turbines in the WRF model

We present the analysis and comparison of the WFPs and the LES-AD results for the single-turbine and two-turbine layouts across the three ABL types. We show the results in two groups: a reference analysis and a joint analysis of wake interactions under the effect of atmospheric stability. First, we present the reference analysis, which consists of the results of the single turbine under neutral atmospheric conditions. The purpose of the reference analysis is to provide a common point of reference with previous studies that tested the Fitch scheme against LES-AD (Archer et al., 2020; Peña et al., 2022) but now with the inclusion of the EWP scheme. Later, we present the analysis of the wake interactions from a simple two-turbine layout under the three ABL types from both WFPs and the LES-AD results.

### 4.1 Reference simulation: Single turbine under neutral atmospheric conditions

Figure 3 shows a vertical cross-section ($x$-$z$ plane) centred on a single turbine for the $\Delta\overline{U}$ and $\Delta\overline{k}$ fields under neutral ABL conditions of the LES-AD simulations. Wind speed differences (Fig. 3a) can be observed behind the rotor area, with absolute differences greater than 4.0 m s$^{-1}$ at about $x = 3000$ m. On the vertical axis, the $\Delta\overline{U}$ values behind the turbine are almost symmetric around the hub height but gradually become skewed to the ground when moving further downstream. The positive values of $\Delta\overline{k}$ (Fig. 3b) at the upper tip of the blade indicate the enhancement of turbulence due to vertical shear. Bellow the turbines hub, there is a small area of TKE sink ($\Delta\overline{k} < 0$ m$^2$ s$^{-2}$). Maximum absolute values of $\Delta\overline{k}$ are found at about $x = 3500$ m.

Figure 4 shows vertical profiles of $\Delta\langle\overline{U}\rangle$ and $\Delta\langle\overline{k}\rangle$ of the Fitch and EWP schemes under different scheme-specific parameters (Table 4) and those of the LES-AD reference. At the turbine's location (Fig. 4a), the profile of wind speed differences from the Fitch simulations compares well with that of the LES-AD. However, the simulations with the EWP scheme are rather sensitive to variations in $\sigma_0$. Using $\sigma_0 = 0.6r_0$ (EWP-0.6) best matches the LES-AD, but other $\sigma_0$ values (EWP-1.0 and EWP-1.7) show significant differences with the LES reference. In Area 2 (Fig. 4b), the LES-AD $\Delta\langle\overline{U}\rangle$ vertical profile indicates a deeper wake than at the turbine location — simulations with the Fitch scheme with all parameter values and EWP-0.6 reproduce this behaviour consistently. Still, there are considerable differences regarding the maximum difference between the LES-AD and



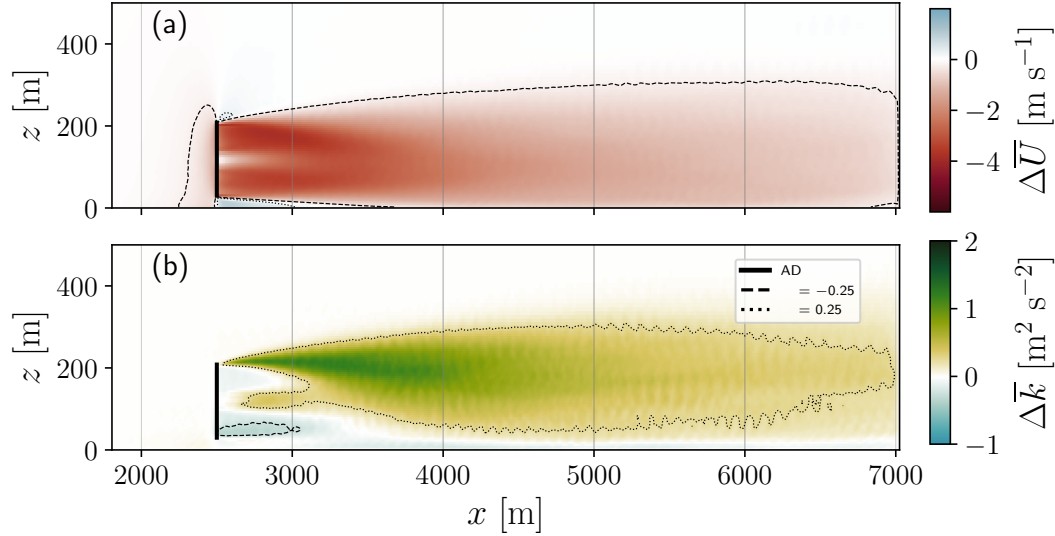

**Figure 3.** Vertical cross-section of the time-averaged differences between the LES with and without a single turbine of (a) wind speed and (b) TKE under neutral atmospheric conditions. The turbine's location is depicted as vertical black lines, and the vertical grey lines show the boundaries of the averaging areas (Fig. 1b) used for the comparison with the PBL simulations. The dashed lines show contours of $\Delta\overline{U} = -0.25$ m s$^{-1}$ and $\Delta\overline{k} = -0.25$ m$^2$ s$^{-2}$, and the dotted lines $\Delta\overline{U} = 0.25$ m s$^{-1}$ and $\Delta\overline{k} = 0.25$ m$^2$ s$^{-2}$ contours.

two EWP runs (EWP-1.0 and EWP-1.7). The LES-AD $\Delta\langle\overline{U}\rangle$ values in Area 3 (Fig. 4c) decrease (in absolute value) with
respect to the upstream areas. None of the Fitch runs capture the wind speed differences as simulated by the LES-AD in this area; however, the EWP-1.0 comes the closest. In the last area (Fig. 4d), EWP-1.0 and EWP-1.7 follow the LES-AD the best, while all Fitch runs show larger differences compared to the LES reference with values up to 0.3 m s$^{-1}$.

Examining $\Delta\langle\overline{k}\rangle$ (Figs. 4e–h), the LES-AD simulations show little TKE production within the area where the turbine is located (Fig. 4e), which is only noticeable at the upper tip of the rotor. This is a consequence of the position of the turbine
within the averaged area where TKE is mostly the same as that at the inflow (see the area delimited between $x = [2000, 3000)$ m from Fig. 3b). The Fitch scheme produces larger TKE values (with differences that depend on the $c_f$ value) than the LES-AD in the area of the turbine location. In Area 2 (Fig. 4f), Fitch-0.25 is the simulation that best matches the LES-AD $\Delta\langle\overline{k}\rangle$ vertical profile and also reproduces well the maximum TKE values around the upper tip of the blades. As we move further down to Area 3 (Fig. 4g), the vertical profiles of Fitch-0.25, Fitch-0.5 and Fitch-0.75 resemble more that of the LES-AD and this similar
behaviour is also seen in the last area (Fig. 4h).

In all investigated areas, the TKE profiles of the EWP scheme are far from those of the LES-AD, with only slight changes, regardless of the scheme-specific parameters used. Note that no TKE source is added to the EWP scheme in its original implementation. When TKE is injected (in the Fitch scheme) at the turbine's location and then advected downstream, the WFP results are comparable to the LES-AD profiles (Figs. 4f–h). Using Fitch-0.0 (without TKE source) and EWP produces
comparable levels of TKE and similar maximum wind speed differences with EWP-0.6.

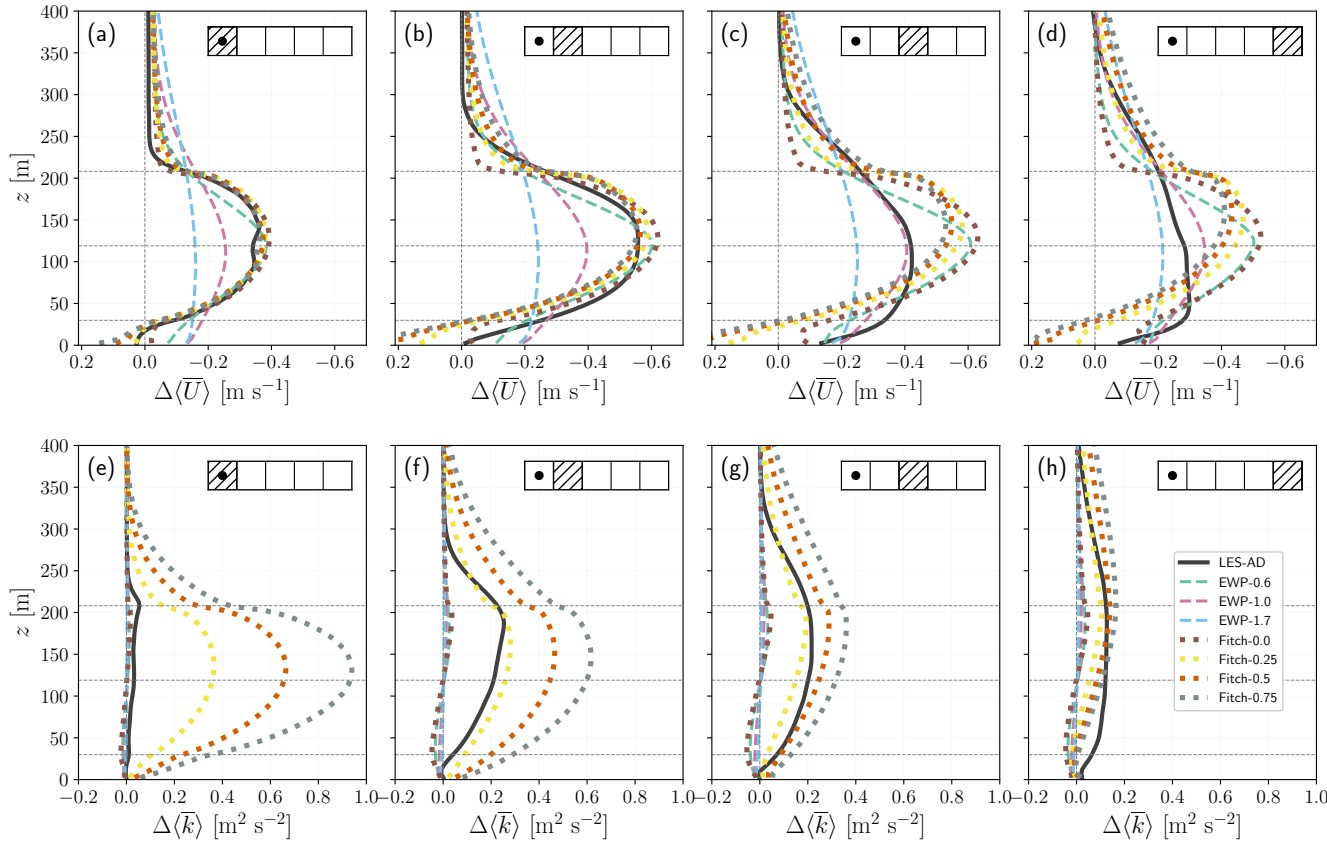

**Figure 4.** Comparison of the vertical profiles of (a–d) wind speed difference ($\Delta\langle\overline{U}\rangle$) and (e–h) TKE difference ($\Delta\langle\overline{k}\rangle$) simulated using EWP (dashed lines) and Fitch (dotted lines) using various scheme-specific parameters (Table 4), and the reference LES-AD (solid black line), for a single turbine under neutral atmospheric conditions. The upper right corner of each panel shows the corresponding averaged areas in Fig. 1b and the position of the turbines (black dots). The heights of the top, bottom, and hub of the rotor are depicted with horizontal grey lines.

## 4.2 Wake and atmospheric stability effects

### 4.2.1 LES-AD cross-sectional fields

Here we move to the two-turbine layout for the three ABL types. Figures 5 and 6 show the horizontal ($x$-$y$ plane at hub height) and vertical ($x$-$z$ plane at the centre of the rotor area) cross sections of the $\Delta\overline{U}$ and $\Delta\overline{k}$ fields from the two-turbine LES-AD simulations under the three ABL types.

As expected, the neutral and stable ABLs (Figs. 5a and e) show larger $\Delta\overline{U}$ than the unstable ABL (Fig. 5c), where the maximum difference can reach $-4$ m s$^{-1}$. The distribution of the LES-AD $\Delta\overline{U}$ under unstable conditions (Fig. 5c) exhibits a wider wake extension (in the $y$-axis) than in stable and neutral conditions (for reference, see the $-0.25$ m s$^{-1}$ contour in Fig. 5a,c and e). In the neutral ABL case (Figs. 5a), the wake is perfectly aligned with the flow direction, whereas under stable





conditions (Fig. 5e), the wake shows a slight deflection. The wake under the three ABL types is confined to the 1-km areas we
delimited to compare with the grid cells of the PBL runs (grey boxes in Fig. 5).

The TKE on the $x$-$y$ plane (Figs. 5b,d and f) reveals a slightly wider wake extension in unstable than in neutral or stable
conditions (see 0.25 m$^2$ s$^{-2}$ contour in Fig. 5b,d and f). The maximum $\Delta \overline{k}$ value across the three ABL types occurs under
unstable conditions after the downstream turbine (Fig. 5d). A barely noticeable TKE sink ($\Delta \overline{k} < 0$ m$^2$ s$^{-2}$) is located behind
the upstream turbine in the three ABL types.

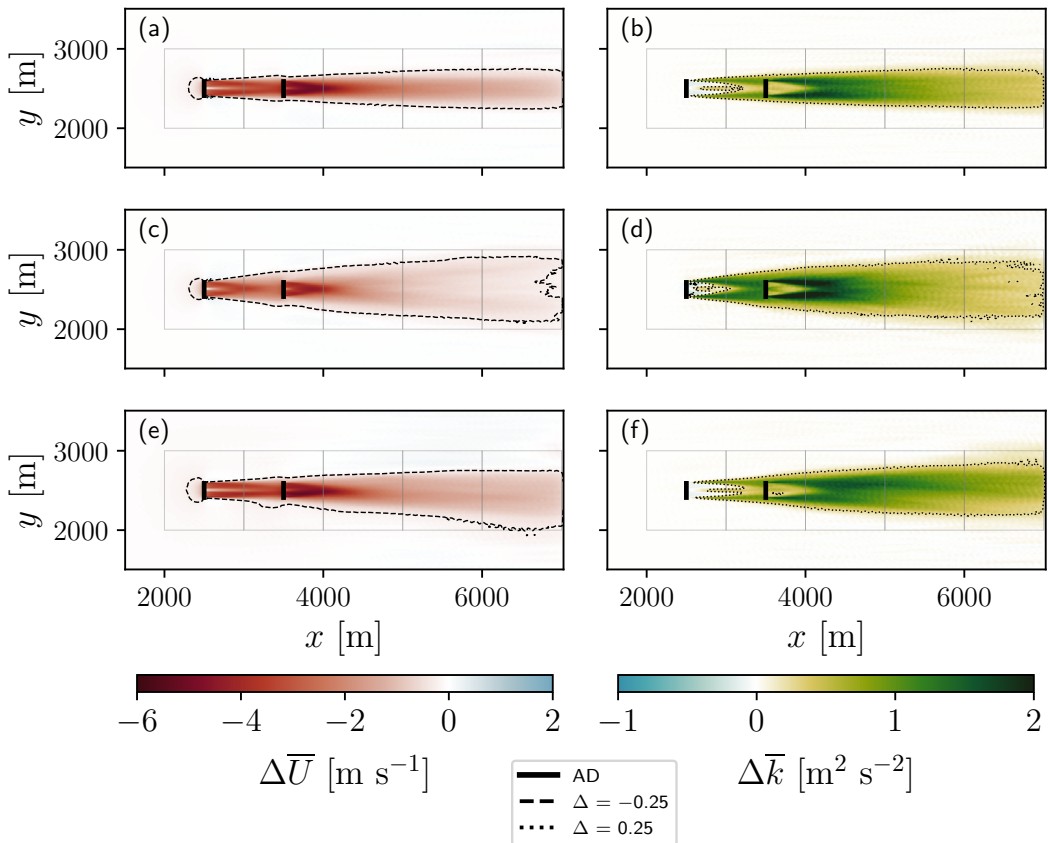

**Figure 5.** Time-averaged horizontal cross sections at hub height of the differences between the simulations without and with two turbines
of wind speed (left panels) and TKE (right panels) using LES-AD under (a–b) neutral, (c–d) unstable, and (e–f) stable ABL. The turbines'
location is depicted as vertical black lines, and the boundaries of the averaging areas used for the comparison with the PBL simulations are
shown in grey. The dashed lines show contours of $\Delta \overline{U} = -0.25$ m s$^{-1}$ and $\Delta \overline{k} = -0.25$ m$^2$ s$^{-2}$, and dotted lines $\Delta \overline{U} = 0.25$ m s$^{-1}$ and $\Delta \overline{k}$
= 0.25 m$^2$ s$^{-2}$ contours.

By analysing the $\Delta \overline{U}$ on the $x$-$z$ plane across the rotor (Figs. 6a,c and e), we identify the blockage effect in the turbines'
induction zone and the speed-up effects around the upper and lower blade tips. The speed-up effects are larger in the stable
ABL than in neutral or unstable ABLs, possibly related to the shallow stable layer (250 m). For the upstream turbine, the



velocity difference (up to $x \approx 3500$ m) and within the rotor area appear vertically symmetric around the hub height under

neutral and unstable conditions (Figs. 6a and 6c, respectively) and points toward the ground under stable conditions (Fig. 6e).

Across the three ABL types, the most significant wind speed difference is behind the downstream turbine, as expected.

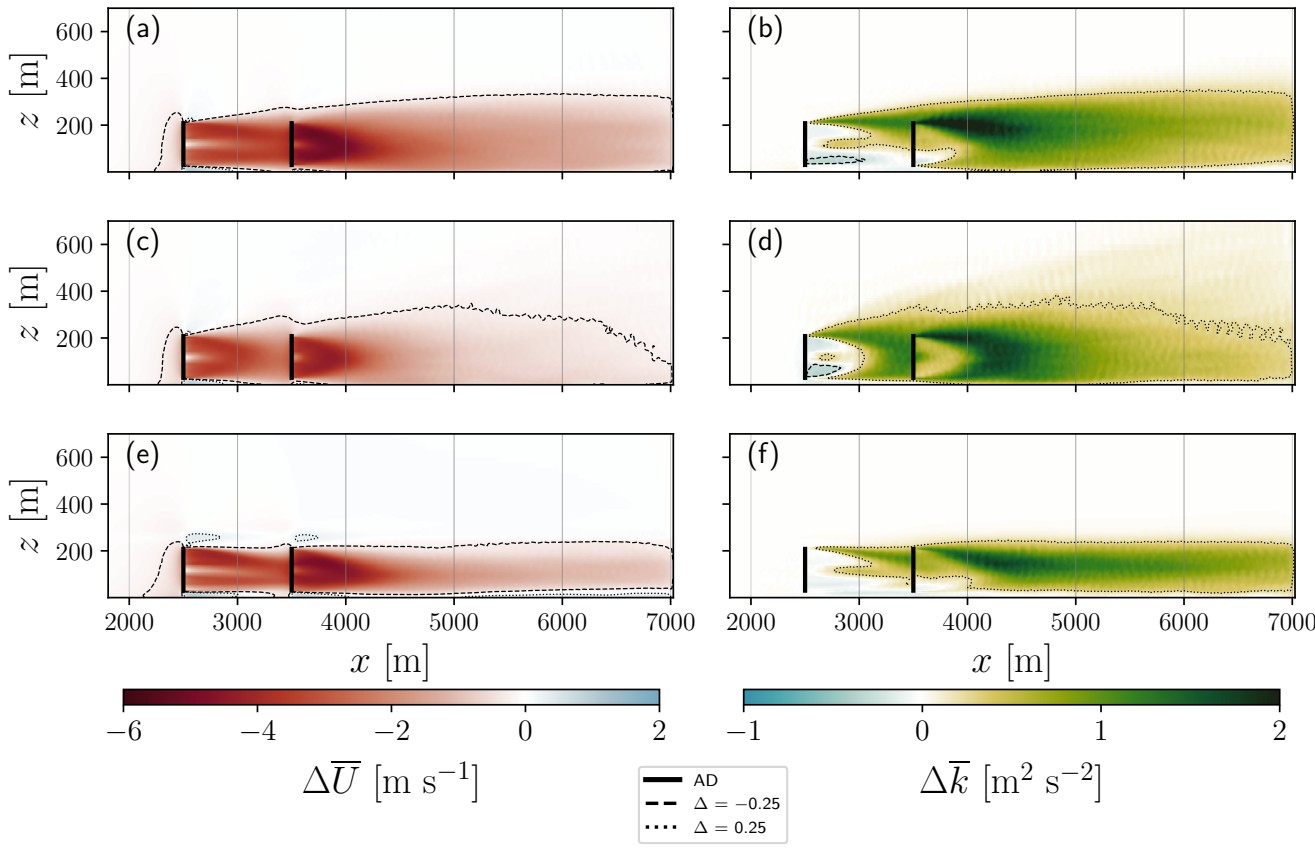

**Figure 6.** As in Fig. 5 but for a $x$-$z$ cross section through the rotor plane.

The TKE $x$-$z$ cross sections under the three ABL types (Figs. 6b,d and f) show the maximum value of $\Delta \overline{k}$ close to $x \approx 4000$

m ($\sim$10.5D downstream of the upstream turbine) and at the height of the turbine's upper tip blade. Generally, the area of large

TKE expands vertically to a greater extent in unstable (Fig. 6d) than in the other two ABL types, with the expansion capped by

the top of the boundary layer (250 m) under stable conditions (Fig. 6f). There is a clear sink of TKE ($\Delta \overline{k} < 0$ m$^2$ s$^{-2}$) below the

turbines' hub height, which varies in extension (on the $x$-axis) depending on the ABL type. The area of TKE decrease reaches

a downstream distance of $\sim$500 m from the upstream turbine in the unstable ABL (Fig. 6d). However, in the neutral and stable

ABLs (Figs. 6b and f), the area reaches $x \approx 1000$ and 1500 m, respectively, from the upstream turbine. Consistently, under the

three ABL types, turbulence develops (positive $\Delta \overline{k}$) at the height of the blade's upper tip.





### 4.2.2 Vertical profiles of wind speed

Figure 7 shows the comparison of the $\Delta\langle\overline{U}\rangle$ profiles of the two WFPs and the LES-AD simulations for the three ABL types. We focus on the three runs using the EWP scheme and two using the Fitch scheme because, based on the single-turbine simulations under neutral conditions, the parameter $c_f$ of the Fitch scheme has less impact on the wind speed field than $\sigma_0$ in the EWP scheme.

Starting at the most upwind location, the stable's LES-AD profile (Fig. 7i) shows a larger wind speed difference than that in the neutral and unstable ABLs. The three ABL types have two $\Delta\langle\overline{U}\rangle$ maxima: above and below the hub height. Contrary to the one-turbine results (Fig. 3a), under neutral conditions (Fig. 7a), the Fitch runs (Fitch-0.0, Fitch-0.25) do not match that well the LES-AD results (neither does the results of EWP-0.6); maximum differences of 0.1 m s$^{-1}$ are found between these profiles. This is partially related to the presence of the downstream turbine. Also, in this area, the EWP-1.0 run compares relatively better with the reference, particularly at hub height and below. Under unstable conditions though, the Fitch runs (Fitch-0.0 and Fitch-0.25) and EWP-0.6 match well the LES-AD $\Delta\langle\overline{U}\rangle$ profile.

Within the area of the downstream turbine, the largest $\Delta\langle\overline{U}\rangle$ (0.92 m s$^{-1}$) is found as expected in the LES-AD under stable atmospheric conditions (Fig. 7j). The WFPs, in contrast, show the largest $\Delta\langle\overline{U}\rangle$ within Area 3. The Fitch-0.0, Fitch-0.25, and EWP-0.6 simulations in Area 2 show the same behaviour as in the upstream turbine: slightly larger $\Delta\langle\overline{U}\rangle$ values than the LES-AD in neutral and stable (Figs. 7b and j) but closer values in unstable conditions (Fig. 7f). Generally, the Fitch-0.0, Fitch-0.25 and EWP-0.6 simulations agree better with the LES-AD $\Delta\langle\overline{U}\rangle$ profile than those of EWP-1.0 and EWP-1.7 in this area.

Within Area 3 (immediately downstream of the downstream turbine), the LES-AD $\Delta\langle\overline{U}\rangle$ values (Figs. 7c,g and k) decrease with respect to the previous area. Also, wind speed differences are larger ($\Delta\langle\overline{U}\rangle < 0$ m s$^{-1}$) above the height of the upper tip than in Areas 1 and 2. None of the Fitch-0.0, Fitch-0.25 and EWP runs capture this "wake recovery" behaviour in Area 3; instead, these WFPs simulate deeper wakes than the LES-AD reference. The EWP-1.0 best matches the reference under the three atmospheric conditions in Area 3, and the EWP-1.7 still simulates weaker wakes than the LES reference.

In the last area, the LES-AD and WFP profiles of $\Delta\langle\overline{U}\rangle$ show a reduction in the wind speed differences ($|\Delta\langle\overline{U}\rangle| \to 0$ m s$^{-1}$) compared to the previous area (Area 3). Except for the EWP-1.7, all the scheme-specific parameters tested for the EWP and Fitch runs produce stronger wakes than the LES-AD profile in the last area under all atmospheric conditions (Figs. 7d,h and l).

To summarise all the previous results, Figure 8 shows the agreement between the WFPs and the LES-AD simulations for each ABL type in the various areas. Again, we use the RMSD within the vertical levels across the rotor area. The WFPs with their scheme-specific parameters generally show larger errors under stable conditions at all locations than under neutral and unstable atmospheric conditions. The differences from the Fitch-0.0, Fitch-0.25, and EWP-0.6 are the smallest ($< 0.01$ m s$^{-1}$) at the turbine's locations (Figs. 8a and 8b, respectively) and progressively become larger downstream. The EWP-1.0 and EWP-1.7, which showed a large RMSD at the turbines' location, now show the smallest errors in Areas 3 and 4 (Figs. 8c and 8d, respectively).

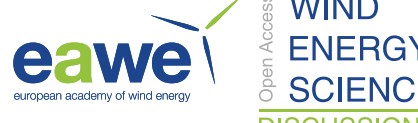

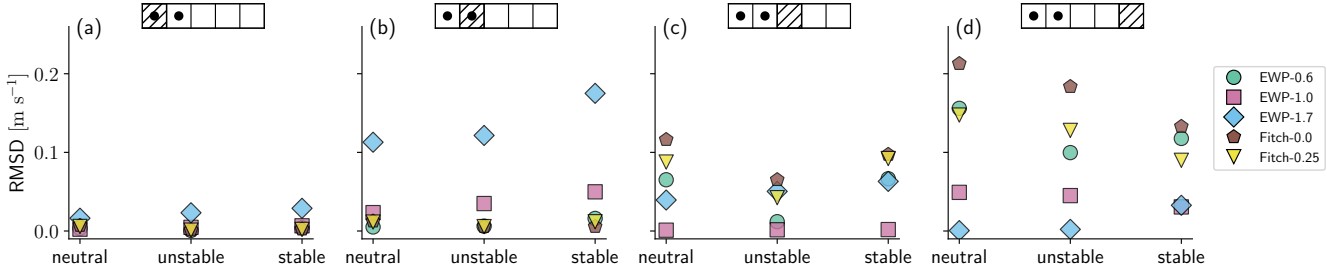

**Figure 7.** Vertical profiles of wind speed difference ($\Delta\langle\overline{U}\rangle$) for the simulations of two aligned turbines under (a–d) neutral, (e–h) unstable, and (i–l) stable atmospheric conditions. The key map above the top panels shows the corresponding averaged areas in Fig. 1b and the turbines' position (black dots). The heights of the top, bottom, and hub of the rotor are depicted with horizontal grey lines.

**Figure 8.** Root mean square difference (RMSD) of wind speed difference $\Delta\langle\overline{U}\rangle$ between the WFPs and the LES-AD simulations across the rotor area, the three ABL types, and study areas (panel's top keymap) for the two-turbine layout.



### 4.2.3 Vertical TKE profiles

Figure 9 shows the vertical profiles of $\Delta\langle\overline{k}\rangle$ for the WFPs and LES-AD simulations under the three ABL types. For simplicity, we only show the LES reference, the three investigated $c_f$ values from the Fitch scheme and EWP-1.7.

**Figure 9.** As in Fig. 7 but for the vertical profiles of TKE difference ($\Delta\langle\overline{k}\rangle$).

Beginning with the upwind-turbine location, the unstable LES-AD profile (Fig. 9b) displays greater $\Delta\langle\overline{k}\rangle$ compared to the neutral and stable ABL, which are nearly negligible except for a small production at the upper rotor tip. Again, this is partially due to the horizontal average procedure (see Figs. 5 and 6). Regarding the WFP results at this location, Fitch-0.25, Fitch-0.5, and Fitch-0.75 overestimate the added TKE by the turbines under all atmospheric conditions.

At the downstream-turbine location (Figs. 9b,f and j), the TKE production from the LES-AD is higher for the unstable (Fig. 9f) than for the other two conditions, as expected. The LES-AD has the highest TKE production at the upper tip of the rotor,



where wind shear drives the TKE production under stable and neutral conditions. This added TKE is more evenly distributed over the rotor area for unstable conditions, resulting in a quasi-Gaussian shape with a maximum around the hub height.

Contrary to the single-turbine simulation under neutral conditions, in Area 2 (see Sect. 4.1), the Fitch-0.25 simulation no longer matches the LES-AD values when simulating two turbines aligned to the flow (Fig. 9b). Instead, the production of TKE from Fitch-0.25 is larger than the LES reference by nearly twice the values. Note that this overestimation is a consequence of the large source of TKE injected at the grid cell where the turbine is located. The Fitch-0.25 also overestimates the reference under stable conditions (Fig. 9j) but provides the best match under unstable conditions (Fig. 9f). Despite this, Fitch-0.25 is the closest to the LES-AD for all ABL types in this area. Fitch-0.5 and Fitch-0.75 greatly overestimate the LES reference.

In Area 3 (Figs. 9c,g and k), we find the maximum TKE production from the LES-AD under all conditions, resulting from the advection of turbulence from the two turbines. In this area, the TKE profiles of the Fitch runs are closer to the reference under all ABL types. Contrary to the results in Area 2, Fitch-0.25 best matches the LES-AD values under neutral conditions (Fig. 9c) and underestimates the LES reference under unstable (Fig. 9g) and stable (Fig. 9k) conditions. For unstable conditions (Fig. 9g), Fitch-0.75 offers the best agreement with the reference in both the maximum and distribution of TKE values than Fitch-0.25. Under the stable ABL, Fitch-0.5 shows good correspondence with the maximum TKE from the LES reference.

In the last area (Figs. 9d,h and l), the LES-AD profiles show the upward vertical transport of TKE that extends the profile beyond 400 m in unstable and neutral conditions. TKE transport under stable conditions is limited by the shallow boundary layer. Moreover, the constraint under stable conditions produces higher TKE values within the rotor area than in the other two ABL types. Concerning the Fitch scheme in the last area, the vertical profiles between Fitch-0.25, Fitch-0.5, and Fitch-0.75 are similar in all ABL types, with maximum differences of $0.1\ \mathrm{m^2\ s^{-2}}$. This set of Fitch runs (Fitch-0.25, Fitch-0.5, and Fitch-0.75) agrees well with the LES-AD values under neutral and unstable conditions (Figs. 9d and h), but misses the TKE distribution under stable conditions.

In general, the Fitch scheme using various parameters reveals that the production of TKE is larger under neutral conditions. The suggested value of $c_f$ from Archer et al. (2020) for the Fitch scheme (Fitch-0.25) appears to be a good fit for the one-turbine layout but does not consistently improve the results for the two-turbine case.

# 5 Discussion

Our results show that the effectiveness of the WFPs varies depending on the variable analysed (wind speed or TKE) and the area of study (at the turbines' site or downstream). Particularly for TKE results, the EWP scheme without an explicit TKE source, and regardless of the $\sigma_0$ values, cannot match that of the LES-AD values. The Fitch scheme using $c_f = 0.25$ provides the closest results to the LES-AD under the range of ABL types studied, but the differences varied depending on the specific area, layout, and ABL type.

Regarding wind speed, the two WFPs offer a distinct and different representation of the wake; they either accurately depict the wake in turbine areas with deviations in the downstream areas, or vice versa. The Fitch scheme with all tested $c_f$ values aligns well with the LES-AD results in the turbine areas but produces deeper wakes in the downstream areas. In contrast,





the EWP's accuracy depends on the $\sigma_0$ value: $\sigma_0 = 0.6r_0$ mirrors the Fitch results, while larger initial length scales (specifi-
cally, $\sigma_0 = 1.0r_0, 1.7r_0$) show better performance downstream but a weaker agreement at the turbine area. Regardless of these
differences, the performance of the WFPs remained relatively consistent across the three ABL types.

Among the three ABL types, the WFPs resemble the least the LES-AD simulations under stable conditions. This is likely
related to the WFPs overlooking the effects of the shallow layer in the stable ABL. The WFPs either misrepresent the wind
speed above the boundary layer (EWP with a $\sigma_0 \geq 1.0r_0$) or do not adequately depict the vertical TKE distribution (Fitch).
Flow acceleration near the surface of the stable ABL is partially captured when TKE is injected in the Fitch scheme with
$c_f \geq 0.25$. However, larger $c_f$ values lead to this acceleration under neutral and unstable conditions (with $c_f \geq 0.5$ in the
unstable ABL; not shown), which are absent in our LES-AD reference. This acceleration is also observed in Fitch simulations
and is absent in horizontally averaged LES results for neutral conditions in previous works (Archer et al., 2020; Peña et al.,
2022).

The results involving the EWP scheme extend previous studies that examined the impacts of $\sigma_0$ for neutral and stable cases
in wind farm simulations (Volker et al., 2015; Badger et al., 2020; Larsén and Fischereit, 2021). Although previous research
calibrated and validated $\sigma_0$ values (within the range $1.5r_0 : 1.9r_0$) and found minor impacts on the wind speed field, our
study further investigates the effect of $\sigma_0$ within a broader range ($\sigma_0 = 0.6r_0, 1.0r_0$, and $1.7r_0$). We find that low values of $\sigma_0$
produce deeper wakes (with deficits concentrated at hub height), and higher values produce more diffuse wakes (the deficit
is distributed vertically). When investigating EWP's TKE production, we find that all types of simulation produce negligible
levels of TKE, a behaviour seen in previous studies when compared with airborne measurements (Larsén and Fischereit, 2021).
This comparison points to a potential area for future improvement in the EWP scheme, which includes an explicit source of
TKE.

Our Fitch results with one turbine under neutral conditions are consistent with those of Archer et al. (2020) and Peña et al.
(2022) using $c_f = 0.25$. Their demonstrated good performance on wind speed in the turbine area and the growth of wind speed
differences in downstream cells align with our findings. However, for the TKE analysis with two-aligned turbines under non-
neutral conditions, the performance varies depending on the analysed location and the ABL type. Under neutral conditions,
$c_f = 0.25$ agrees well with the reference within the downstream turbine areas for single and two-turbine layouts. But for the
two-turbine layout under unstable and stable conditions, $c_f = 0.5$ and $0.75$ seem better choices. These changes in performance
depending on the location and atmospheric conditions highlight the limitations of current WFPs and underscore the need for
further research to improve their parameters.

Our study uses LES-derived data to assess the accuracy of the TKE and wind speed representations in both WPFs, provid-
ing the means of validation with limitations. Given the distinctive temporal and spatial length scales used when employing
PBL approximations and LES, achieving identical inflow profiles remains a significant challenge and, consequently, a perfect
resemblance should not be expected. Developing these conditions in the WRF model requires running the model under differ-
ent combinations of boundary conditions, initial profile soundings and heat fluxes. Testing each combination is an expensive
procedure. In addition, long simulations are needed to reach a point when the inertial flow oscillations have decayed while





considering that the boundary layer might continuously grow. Here, we focused on matching similar conditions at hub height and found good agreement within the rotor area.

Despite these efforts, there were still differences in inflow conditions between the PBL and LES simulations. One of these differences is the higher production of TKE in the LESs compared to that of the PBL scheme. These differences are more prominent near the surface and extend vertically to hub height, which could potentially modify the wind speed deficit due to wake. Furthermore, high TKE values translate into high wake recovery rates (more mixing). Besides the different temporal and spatial scales between the LES and PBL frameworks, the higher background TKE and that added from the turbines could

contribute to faster wind speed wake decay (or mixing) in the LES simulations, explaining some of the differences found.

The dependence of the WFPs performance in the various wake regions might be related to the choice of the MYNN scheme. Among the various PBLs available in the WRF model, MYNN is currently the only scheme that can be coupled with the Fitch scheme, primarily due to the TKE advection feature required by this WFP. Consequently, we use the EWP with this PBL for consistency, although it is not dependent on this feature. Using different PBL schemes together with WFPs could potentially

provide a more representative comparison with the LES-AD. Particularly promising are the new PBL schemes that introduce 3D turbulence features (Eghdami et al., 2022), which theoretically could offer a more accurate representation than traditional 1D PBL schemes.

## 6   Conclusions

The use of WFPs in mesoscale models plays an important role in understanding the response of wind farms to atmospheric

flow. This work evaluated two WFPs in the WRF model, Fitch and EWP, under three distinct atmospheric stability conditions using the WRF model in LES mode as the reference. The evaluation reveals in detail the manner and consequences of using parameterizations that miss the intricacies of the flow provided by the LES reference.

Compared to the LES reference, the performance of the two WFPs evaluated depends heavily on the area of interest (range of downwind areas) and the variable analysed:

– The WFPs can depict the vertical profiles of the wind speed differences in two distinct scenarios: (1) good agreement in the grid cells where the wind turbines are located but deeper wakes downstream, *or* (2) good agreement in the grid cells in the downstream areas but weaker wakes at the turbine locations. Specifically, the Fitch scheme aligns with the first scenario. Meanwhile, the behaviour of the EWP depends on its parameter value, allowing it to emulate either of the two described behaviours.

– To match the reference TKE values, the WFPs need an explicit source of TKE. When this source is included, there is good agreement with the reference TKE values in the downstream grid cells and an overestimation in the grid cells that contain the turbines. This TKE source should also be adjusted based on the atmospheric stability conditions to improve accuracy.





The variations in performance are consistently observed under the three atmospheric stability conditions, with the WFPs
resembling the least in stable conditions. Various factors may influence the performance patterns listed above, including the
specific PBL parameterization used. Recognising these variations, our study identifies potential areas for improving the WFPs.
Future studies may expand on this work by exploring alternative PBL schemes and considering factors such as varied turbine
layout, spacing, and farm sizes.

In their current state, WFPs only offer a partial view of wake dynamics compared to the LES reference. Because of this, it
is essential that the WRF model users understand how the performance of their chosen WFPs and the setup may influence the
reliability of the results for a given use case. Assessing performance across downstream cells can help understand wind farms'
effects on wind resources and nearby farms, and achieving accurate results at the turbine location can lead to more refined and
effective power modelling. Ultimately, these insights emphasise the importance of using WFPs appropriately based on their
intended application.

*Code and data availability.* The simulation results are too large to archive. Therefore, we will provide the necessary files to replicate the
simulations once the article is accepted for publication.

The WRF model is an open source code available at https://github.com/wrf-model/WRF. The Fitch scheme is integrated into WRF's main
public releases, and the EWP scheme and the AD implementation will soon be integrated into the official WRF repository. In the meantime,
the EWP's source code is provided upon request.

**Appendix A: Definition of the parameters in the wind farm parameterization**

The WFPs only have a user-specified parameter: The TKE correction factor ($c_f$) for the Fitch scheme and the initial length
scale ($\sigma_0$) for the EWP scheme. In the Fitch scheme, the momentum sink from the turbines is related to the turbine's trust
coefficient and the model's grid area. The deceleration is applied to the model's velocity tendencies ($u$ and $v$) proportionally
to the intersecting area ($A_m$) between the vertical levels of the model and the swept rotor area of the turbine (Eqs. 8–10 from
Fitch et al., 2012). Similarly, sources of TKE are computed using:

$$\frac{\partial \text{TKE}_m}{\partial t} = \frac{1}{2} \frac{N_\text{t}}{\Delta x^2} \frac{A_m C_\text{TKE} U_m^3}{z_{m+1} - z_m}, \tag{A1}$$

where $N_\text{t}$ is the number of turbines within the grid cell, $\Delta x$ is the model's grid spacing, and $U_m$ and $z_m$ are the wind speed and
height at each vertical level ($m$) within the rotor area, respectively. $C_\text{TKE}$ is the difference between the thrust $C_T$ and power
$C_P$ coefficients (for a given wind speed). Archer et al. (2020) suggested the addition of a correction factor ($c_f$) for the $C_\text{TKE}$
coefficient:

$$C_\text{TKE} = c_f(C_T - C_P). \tag{A2}$$

The EWP scheme explicitly resolves the deficit in the far wake exerted by a turbine or wind farm. The scheme uses classical
wake theory based on the solution of a parabolic equation where horizontal advection and diffusion of turbulence dominate.



The solution describes the velocity deficit profile in a position $x$ in the far wake (Tennekes and Lumley, 1972):

$$\overline{u}_d(x,y,z) = \overline{u}_s \exp\left[\frac{1}{2}\left(\frac{z-h}{\sigma}\right)^2 - \frac{1}{2}\left(\frac{y}{\sigma}\right)^2\right] \tag{A3}$$

where $\overline{u}_d$ represents the velocity deficit, $\overline{u}_s$ the maximum velocity deficit, $h$ hub height, and $\sigma$ the length scale. The distribution of $\overline{u}_d$ (in the $y$-$z$ plane) at a distance $x$ is given by the length scale $\sigma$ given by

$$\sigma^2 = \frac{2K_h}{\overline{U}_h}x + \sigma_0^2, \tag{A4}$$

where $K_h$ and $\overline{U}_h$ are the coefficient of diffusion of turbulence of momentum at hub height $h$ and the advection velocity at the
same height. $\sigma_0$ represents the initial length scale, which is a fraction of the turbine's rotor radius $r_0$. The unresolved far wake is represented by equation A3 and accounts for the near wake with the initial length scale. The volume-averaged velocity deficit equations (Eqs. 13–15 from Volker et al., 2015) are found by equating the deficit profile to the turbine's thrust and integrating over model's grid cell area:

$$\langle \overline{f}_m \rangle = -N_t \sqrt{\frac{\pi}{8}} \frac{r_0^2 C_T \overline{U}_h^2}{\Delta x^2 \sigma_e} \exp\left[-\frac{1}{2}\left(\frac{z_m-h}{\sigma_e}\right)^2\right], \tag{A5}$$

where $\sigma_e$ is the effective length scale that accounts for the wake expansion over a downstream distance $L$ within the grid cell:

$$\sigma_e = \frac{1}{L}\int_0^L \sigma \,\mathrm{d}x = \frac{\overline{U}_h^2}{3K_m L}\left[\left(\frac{2K_m}{\overline{U}_h^2}L + \sigma_0^2\right)^{\frac{3}{2}} - \sigma_0^3\right] \tag{A6}$$

*Author contributions.* OGS wrote the first draft, carried out the experiments and analysed the results. All authors participated in the conceptualisation, analysis design and the manuscript's writing and editing.

*Competing interests.* AH is a member of the editorial board of Wind Energy Science. The authors declare that they have no other competing
interests to declare.

*Acknowledgements.* OGS and JB acknowledge the support of the European Union Horizon 2020 research and innovation program under grant agreement no. 861291 as part of the Train2Wind Marie Skłodowska-Curie ITN (https://www.train2wind.eu/). AP and AH were funded by the Independent Research Fund Denmark through the 'Multi-scale Atmospheric Modeling Above the Seas' (MAMAS) project. The authors thank Jeffrey Mirocha at Lawrence Livermore National Laboratory for access to the actuator disc implementation and helpful dis-
cussions. We also acknowledge the computational resources provided on the Sophia HPC Cluster at the Technical University of Denmark, DOI: 10.57940/FAFC-6M81.





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
