# Peer review of "Evaluation of wind farm parameterizations in the WRF model under different atmospheric stability conditions with high-resolution wake simulations"

_Wind Energy Science, 2023_

## Author Comment (AC1)

**Reply to reviews – WES-2023-124**

Oscar García-Santiago, Andrea N. Hahmann, Jake Badger, and Alfredo Peña

08-01-2024

**Reviewer #1**

We appreciate your positive feedback on our manuscript. The details of these revisions are described below, and for clarity, the original comments from the reviewers are presented in black, while our corresponding responses are highlighted in blue:

This work compares the performance of two wind farm parameterizations in WRF for their skills at wake representation versus a large-eddy simulation. The analysis is thorough, relevant, and considers single and multiple turbine layouts and each atmospheric stability regime. The paper is very well written with compelling figures. This work is recommended for publication after the following minor suggestions and corrections are addressed.

1. The results section would benefit from the addition of numerical values throughout to help the reader understand what the authors mean in statements such as "nearly negligible," "larger errors," "considerable differences," and "best agreement." A few sentences that would benefit from such numerical additions are explicitly stated below, but a comprehensive review of the manuscript in support of the addition of numerical findings is recommended.
   Thank you for pointing this out. We have revised the manuscript in the manner suggested and replaced such statements with quantitative ones. In a number of instances, we have used a relative difference (i.e. $(experiment - reference)/reference$) with the reference being the LES runs. Specific changes are highlighted in the diff text document.

2. In the discussion section, it would be of interest to tie the results to wind generation by employing the reference turbine's power curve. For example, it would be interesting to see what the sensitivities in generation or capacity factor estimates over the analysis hours are relative to the sensitivities noted for each WFP, correction factor, and length scale.
   As suggested by the reviewer, we have calculated (post-processed) the relative power difference (%) of the two-turbine simulations (Table 2). Since the power generation is a function of the wind speed, changes in the $\sigma_0$ parameter from the EWP have more impact in the extracted power than the $cf$ parameter from the Fitch WFP that modifies the amount of TKE injected in the

atmosphere. The difference between the minimum and maximum relative power difference obtained from the various $\sigma_0$ tested, are 8.5, 13.1, and 15.4% under unstable, neutral, and stable atmospheric conditions respectively. We have added a paragraph in the Discussion section where we provide a discussion on the power sensitivity from the WFP parameters and their implications.

Table 2: Relative power difference (%) of the two-turbine layout for each simulation under different atmospheric stability regimes. In all cases, power is computed from the cell-averaged wind speed ($\langle \overline{U} \rangle$) at hub height and the reference is the LES runs.

| Run | Atmospheric stability | | |
|---|---|---|---|
| | Unstable | Neutral | Stable |
| EWP-0.6 | 2.74 | $-3.90$ | $-3.21$ |
| EWP-1.0 | 6.32 | 3.45 | 5.52 |
| EWP-1.7 | 11.21 | 9.17 | 12.19 |
| Fitch-0.25 | 0.94 | $-3.0$ | $-1.97$ |
| Fitch-0.0 | $-0.21$ | $-4.34$ | $-2.25$ |

3. Line 124: "stated" should be "state"
   Fixed as suggested.

4. Line 133: Recommend adding some discussion as to why MYNN was selected as the PBL scheme for this analysis, along with speculation based on the literature on how an alternate scheme might impact your analysis and results.

   Our choice of the MYNN PBL scheme is primarily driven by the fact that the TKE advection is activated when using this scheme. This is not the case for other PBL schemes that include TKE. It allows for the transport of the explicit TKE source from the turbine from one grid point to its neighbours and for the TKE to be "remembered" from one time step to the next . Consequently, most wind farm simulations using the WRF model reported in the literature have utilized this PBL scheme. To our knowledge, only a few studies (Peña et al., 2023; Rybchuk et al., 2022) have employed the Fitch scheme with alternative PBLs, such as the NCAR 3DPBL (Juliano et al., 2022). However, the 3DPBL scheme has not been merged yet with the community-open WRF version. The EEPS PBL (Zhang et al., 2020) scheme also has advection of TKE, but we have not found an study of using the Fitch WFP and this PBL.

   In terms of atmospheric stability, using a different PBL scheme than the MYNN could be advantageous, as some PBL schemes demonstrate better performance in modeling wind speed for specific regions. For example, Draxl et al. (2014) suggested using the MYJ scheme for stable conditions, ACM2 for neutral, and YSU for unstable conditions in Northern Europe. These alternatives might offer improved wake representation under those atmospheric conditions. However our results indicate that the differences from the reference are larger when comparing the wake regions than when comparing the atmospheric stability cases. Actual wind farm simulations with the WFPs along different PBL schemes are needed to investigate the wake regions.

   We have acknowledged the use of the MYNN scheme in our methodology section and expanded

the discussion to include the implications of employing different PBL schemes.

5. Line 217: Suggest adding numerical values throughout this paragraph to help the reader understand what is meant by "significant differences" and "considerable differences".
   As suggested, we have added supporting numerical values to these sentences

6. Line 259: Should the word "and" be removed from this line?
   Removed as suggested.

7. Line 308: "Fitch-0.25, Fitch-0.5, and Fitch-0.75 overestimate the added TKE by the turbines under all atmospheric conditions" By how much do they overestimate?
   Numerical values are added to support the statement. Now it reads: *Fitch-0.25, Fitch-0.5, and Fitch-0.75 overestimate the added TKE by 0.24, 0.50, and 0.74 $m^2$ $s^{-2}$, respectively, under all atmospheric conditions at hub height.*

8. Line 318: "Fitch-0.5 and Fitch-0.75 greatly overestimate the LES reference" Again, by how much?
   As suggested, we have modified the sentence. Now it reads: *Fitch-0.5 and Fitch-0.75 greatly overestimate the LES reference at hub height by 209% and 326%, respectively.*

**References**

Caroline Draxl, Andrea N. Hahmann, Alfredo Peña, and Gregor Giebel. Evaluating winds and vertical wind shear from Weather Research and Forecasting model forecasts using seven planetary boundary layer schemes. *Wind Energy*, 17(1):39–55, 2014. ISSN 1099-1824. doi: 10.1002/we.1555.

Timothy W. Juliano, Branko Kosović, Pedro A. Jiménez, Masih Eghdami, Sue Ellen Haupt, and Alberto Martilli. "Gray Zone" Simulations Using a Three-Dimensional Planetary Boundary Layer Parameterization in the Weather Research and Forecasting Model. *Mon. Wea. Rev.*, 150(7):1585–1619, July 2022. ISSN 1520-0493, 0027-0644. doi: 10.1175/MWR-D-21-0164.1.

Alfredo Peña, Oscar García-Santiago, Branko Kosović, Jeffrey D. Mirocha, and Timothy W. Juliano. Can we yet do a fairer and more complete validation of wind farm parametrizations in the WRF model? In *Journal of Physics: Conference Series*, volume 2505, page 012024. IOP Publishing, May 2023. doi: 10.1088/1742-6596/2505/1/012024.

Alex Rybchuk, Timothy W. Juliano, Julie K. Lundquist, David Rosencrans, Nicola Bodini, and Mike Optis. The sensitivity of the fitch wind farm parameterization to a three-dimensional planetary boundary layer scheme. *Wind Energy Science*, 7(5):2085–2098, October 2022. ISSN 2366-7443. doi: 10.5194/wes-7-2085-2022.

Chunxi Zhang, Yuqing Wang, and Ming Xue. Evaluation of an E–$\epsilon$ and Three Other Boundary Layer Parameterization Schemes in the WRF Model over the Southeast Pacific and the Southern Great Plains. *Mon. Wea. Rev.*, 148(3):1121–1145, February 2020. ISSN 1520-0493, 0027-0644. doi: 10.1175/MWR-D-19-0084.1.

---

## Author Comment (AC2)

**Reply to reviews – WES-2023-124**

Oscar García-Santiago, Andrea N. Hahmann, Jake Badger, and Alfredo Peña

08-01-2024

**Reviewer #2**

Thank you for the positive evaluation of our manuscript. Your comments are in agreement with those of referee #1 and led to a slight revision of the paper. The specific changes are outlined below. In our response, the reviewers' original comments are in black along with an English translation, our responses and additions to the manuscript are in blue, and deleted text from the manuscript in strike-through red:

本手稿的作者将 WRF 模型中两种广泛使用的 WFP 与使用同一模型执行的涡轮机尾流的 LES 进行了比较。评估了 Fitch 方案和显式尾流参数化在中性、不稳定和稳定大气稳定性条件下表示双涡轮风电场布局中的风速和湍流动能 (TKE) 的能力。本文讨论的主题很有趣，而且论文结构良好且易于理解。我衷心祝贺这篇手稿的作者，我认为在稍作修改后应该考虑出版。

*Translation*: The authors of this manuscript compared two widely used WFPs in WRF models with LES of turbine wakes performed using the same model. The Fitch scheme and explicit wake parameterization are evaluated for their ability to represent wind speed and turbulent kinetic energy (TKE) in a twin-turbine wind farm layout under neutral, unsteady and steady atmospheric stability conditions. The topic discussed in this article is interesting and the paper is well structured and easy to understand. I sincerely congratulate the authors of this manuscript, which I believe should be considered for publication after minor revisions.

我认为，这些手稿值得在以下问题上进行修改。

*Translation*: In my opinion, these manuscripts deserve revision on the following comments.

Q1 . 第 120 行。请添加一些解释，说明为什么选择 2% 作为阈值。

*Translation*: Q1. Line 120. Please add some explanation as to why 2% was chosen as the threshold.

The ideal criteria would be almost "no change", but due to the inherent unsteadiness of LES, and considering the boundary layer growth and inertial oscillations, we use a larger threshold. We follow the criteria of 2% threshold following the work of Maas and Raasch (2022) for large offshore

wind farm simulations in LES under different atmospheric stability conditions. The reference has been added to the main text.

Q2 第 215-225 行应提供更多定量结果，尤其是在描述显着差异和相当大差异时。

*Translation*: Q2 lines 215-225 More quantitative results should be provided, especially when describing significant differences and considerable differences.

As suggested, we have added supporting numerical values to these sentences

Q3 . 第 320-325 行文本中应提供定量值，以显示最佳一致性和良好对应关系。*Translation*: Q3. Lines 320-325 Quantitative values should be provided in the text to show optimal consistency and good correspondence.

As suggested, we have made the following changes in the manuscript:  Fitch-0.75 offers the best agreement with the reference  hub-height $\Delta\langle\overline{k}\rangle$ with overestimations of 8.9% and 4.8% under unstable and stable conditions, respectively.

**References**

Oliver Maas and Siegfried Raasch. Wake properties and power output of very large wind farms for different meteorological conditions and turbine spacings: A large-eddy simulation case study for the German Bight. *Wind Energy Science*, 7(2):715–739, March 2022. ISSN 2366-7451. doi: 10.5194/ wes-7-715-2022.